# Using SenseMaker® to Understand the Prioritization of Self-Care and Mental Health of Minoritized Engineering Students during the 2020 Global Pandemic

**Racheida S. Lewis** [1,*] , **Trina Fletcher** [2,*], **Animesh Paul** [1], **Diane Abdullah** [2] **and Zaniyah V. Sealey** [1]

1   Engineering Education Transformations Institute (EETI), University of Georgia, Athens, GA 30602, USA
2   School of Universal Computing, Construction & Engineering Education (SUCCEED),
    Florida International University, Miami, FL 33174, USA
*   Correspondence: rslewis@uga.edu (R.S.L.); trfletch@fiu.edu (T.F.)

**Abstract:** The 2020 global pandemic caused by COVID-19 forced higher education institutions to immediately stop face-to-face teaching and transition to virtual instruction. This transition has been difficult for engineering education, which has strong hardware, software, and practical/laboratory components, and has further exacerbated the personal and professional experiences of minoritized students in engineering. This study sought to answer the following overarching research question: *How has the abrupt transition to online instruction due to COVID-19 affected students traditionally underrepresented in engineering?* The abrupt transition for minoritized students and their decision to prioritize their mental health was further explored to answer the following: (1) How many minoritized students chose to prioritize their mental health? (2) How do minoritized students describe their experiences and choices to prioritize (or not) their mental health? Using SenseMaker, participants shared stories using the following prompt: *Imagine you are chatting with a friend or family member about the evolving COVID-19 crisis. Tell them about something you have experienced recently as an engineering student.* After completing their narrative, a series of triadic, dyadic, and sentiment-based multiple-choice questions were presented. Student responses varied, including positive experiences, which resulted in a strong prioritization, while others had negative experiences resulting in varied prioritizations. Some students chose to prioritize their mental health to remain mentally and emotionally healthy; some with negative experiences abandoned self-care strategies in order to tend to the needs of academics and family. Participants' decisions to prioritize their mental health were not monolithic.

**Keywords:** self-care; mental health; engineering; COVID-19; minoritized students; global pandemic

## 1. Introduction

The COVID-19 global pandemic, which hit the United States in early spring 2020, forced higher education institutions to immediately stop face-to-face teaching and transition to virtual instruction. As of February 2021, the pandemic is still in full force, placing most of the nation's institutions in a majority remote learning and teaching environment for the third semester straight, fourth if counting summer 2020 [1]. The impacts of the pandemic on higher education programs have led to the closure of academic programs, faculty and staff layoffs, and loss of income from on-campus dining and housing, which has led to an overall negative impact on budgets and fiscal outlooks [2]. STEM academic programs that have extensive requirements linked to software, hardware (i.e., laptops, desktop computers), and labs, have taken a hard hit in meeting those demands remotely [3]. When focusing on engineering education, there have been strong efforts to broaden the participation of minoritized students including women, first-generation students, and low socioeconomic status students (LSESs) [4]. Preliminary research covering the pandemic's impact on engineering students is finding that many of them, especially minoritized students, are

being negatively impacted, personally and professionally, due to the abrupt and ongoing nature of the pandemic [5,6].

## 1.1. Background

The abrupt transition to virtual learning has been difficult for many students taking STEM courses, particularly engineering, which has a strong practical/laboratory component [7]. Additionally, the ongoing implications and uncertainty surrounding the COVID-19 pandemic have further exacerbated the personal and professional experiences of minoritized students in engineering. To better understand the impact of the pandemic on minoritized students' experiences, the research study was guided by the following overarching research question: *How has the abrupt transition to online instruction due to COVID-19 affected students traditionally underrepresented in engineering*? In the process of analyzing results, a significant finding arose around the area of self-care. While public health officials have urged people to practice self-care and to take care of their mental health, to the point that several mental health services have been provided at a discounted rate or for free, challenges still exist. Additionally, despite the national call and efforts to promote self-care, our results found that very few minoritized engineering students felt they had the privilege to do so.

One could hypothesize that the recommendations consistently communicated by the Centers for Disease Control and Prevention (CDC) to practice social distancing and maintain a minimum of 6 feet apart from those you do not reside with has played a role in this. People are naturally social beings; therefore, adhering to the advice to socially isolate coupled with the rigor of engineering and the added pressure of finishing a once-in-person semester, virtually, could be an influential cause of increased stress on minoritized students. In order to obtain a more concrete understanding of this phenomenon, this study aimed to answer the following research questions:

(1) How many minoritized students chose to prioritize their mental health?
(2) How do minoritized students describe their experiences and choices to prioritize (or not) their mental health?

## 1.2. Literature Review

The COVID-19 pandemic has highlighted the importance of mental health for everyone. One of the populations highly affected by the pandemic and its restrictions is college students [8]. Numerous studies illustrate the challenges faced during the pandemic for both teachers and students. Even before the pandemic, engineering undergraduate students expressed a negative association between poor mental health and pursuing engineering [9]. There have been studies specifically regarding student mental health in higher education during the pandemic. For instance, researchers at a large university in Texas conducted semi-structured interviews to discover what coping skills students were using to address their stress related to the pandemic. They revealed that students were coping not by seeking support from mental-health professionals but by self-management and courting help from family and friends [8]. Regardless of growing nationwide attention, undergraduate engineering programs have minimal research and experience in addressing mental health concerns [10]. A study to examine what methods kept STEM students actively engaged in courses suggests that teachers of online courses drive student participation instead of assuming that desired engagement will come from the students themselves [11]. Mental health is a potent forecaster of educational and workplace productivity [12]; if engineering departments aim to equip students to succeed, they must address and support their mental health [12].

Research has described a stress culture in engineering, where undergraduate students expect or deem high-stress levels and poor mental health necessary for success [10]. The relationship between high stress and poor mental health is quite upsetting, provided the rate of stress increased for college students during the pandemic. Previous studies have illustrated multiple stressors for college students, including family, romantic, peer, and faculty relationships, lack of resources, expectations, academics, environment, diversity,

and transitions [13]. Despite acknowledging the existence of mental health, students do not seek treatment due to lack of time, privacy, stigma, lack of emotional openness, and financial constraints [14]. A similar study regarding the Headspace app illustrated that stress and anxiety could negatively impact student experience, academic performance, and retention [1]. That same study also observed that the mental health crisis is crucial in engineering and students may correlate pursuing engineering with poor mental health. A separate study supports that graduate students in STEM are at elevated risk for mental health issues, particularly women [1,3]. These studies assert that a culture of stress has consequential implications for the engineering ecosystem, and it is an obstacle for students to not only join but also strive in engineering. One can infer that the additional stressors of the pandemic can further aggravate these already existing mental health challenges. Additional stressors may consist of issues with internet access, limited contact with peers, the short amount of time to resolve issues, and the uncertainty of the duration of distance learning [15]. Some of these stressors were disruptions to what was once readily accessible to the student, such as changes to students' living arrangements and the financial resources available [16]. The lack of feeling connected to peers and mentors (due to social distancing requirements) may contribute to a student's hesitation to prioritize their mental health or seek support to manage any challenges they may be facing.

Studies conducted by Chang et al. and Wang et al. focused on the impact COVID-19 had on college students at the graduate (both) and undergraduate (Wang et al.) levels. The adaptations higher education had to undertake during the global pandemic are nationally known in the U.S. These studies found that students had increased severity of depression and anxiety as a result of living through the pandemic, and this was exacerbated for students identifying as women, low-SES students, international students, and students with disabilities [17,18]. This resulted in delaying graduation, having negative perceptions of future job searches, and experiencing symptoms of depression and anxiety several days a week on average [17,18]. It is important to note that while these studies explored the impact of COVID-19 on the mental health of college students broadly, both fail to explore this impact through an intersectional lens of race. Thus, this paper aims to add to this discussion with the understanding that in the U.S., COVID-19 disproportionally impacted people of color, and thus, the impact on their mental health is also of paramount focus. Due to the lack of support for mental health both in the field of engineering and in communities of minoritized populations prior to COVID-19, as well as the clear exacerbation of these stressors since the onset of the pandemic, this paper aims to discuss how these stressors have impacted minoritized engineering students and the steps taken by students in response to these stressors. Section 1.3 will highlight the theoretical framing used to develop the SenseMaker platform that enabled us to capture perceptions of mental health during the pandemic.

### 1.3. SenseMaker Platform

We employed a mixed methods research tool called SenseMaker which "combines first-hand narratives with the statistical authority of quantitative data" [19]. A signification framework is built into the software by researchers in order to "guide the inquiry, solicit micro-narratives, and clarify interpretations which extract associated meanings of the narratives from participants" [19]. The sensemaking process takes place when participants answer signifier questions relative to the narrative they shared. Answers to the signifiers create numerical coordinates that are associated with the participant's narrative and can be compared to other participants within the platform. This allows for both qualitative and quantitative data to be viewed and analyzed in parallel.

It is important that we make the distinction between SenseMaker analysis and narrative analysis. Narrative inquiry focuses on the collection and analysis of stories to understand an experience. While this is a strength of this methodological approach, it is also its weakness for the research team, particularly for a large number of participants, given the time it takes to collect, transcribe, and analyze data [20]. As narrative studies of minoritized students increase in popularity as a way to explore the experiences of these

groups [21–23], the SenseMaker approach allows us to collect rich data akin to what we could collect using narrative analysis and combine these narratives with detailed quantitative data, while also reaching a wide range of participants and avoiding the issue of "small N" that is associated with these groups when using quantitative methods [24].

Theoretical Framework Used to Develop the SenseMaker Platform

Astin's Theory of Student Involvement focuses on institutional outcomes in relation to student development as a result of their level of involvement and engagement. There are three components of focus: (1) student inputs (i.e., background, demographics, etc.), (2) the student's surroundings or environment (i.e., holistic set of experiences while in college), and (3) student outcomes (i.e., who they are, what they believe once they leave the institution). In addition to the three major components, Astin developed five (5) assumptions linked to his theory of student involvement. First, there is mental and physical energy associated with being and staying involved as a student. Second, the involvement is ongoing and will not be the same for all students. Third, there are both quantitative and qualitative experiences and components linked to students. Fourth, student outcomes from their involvement are correlated with how much they were actually involved and how those experiences were. Finally, the fifth assumption is that a student's academic outcomes (i.e., persistence, performance, etc.) are directly linked to their level of involvement and engagement at the institution [25,26]. When referring to the three components of focus presented by Astin, our study incorporates all three. Student inputs are in line with our focus on minoritized students in engineering which covers their race/ethnic backgrounds and their academic programs, which is key to this study. Secondly, the student's environments were impacted by COVID-19, and this study dives into how their environments were impacted. Third, this study captures how students feel their academic outcomes will be impacted as a result of COVID-19. Most importantly, and particularly, our study researches all three aspects with mental health as a key focus.

To determine the signifiers for the SenseMaker survey, we researched and employed the theory of thriving. Tobias [27] defines thriving as an attempt to "highlight the relationship between organizational enlightenment (the thriving organization) and personal maturity (the thriving person)" with the goal of strengthening organizational cultures. We chose this framework specifically for Tobias' definition of a thriving person: "a microcosm of the dilemmas, the tensions, the risks, and the opportunities inherent in 'rearing' a culture" (p. 4). This idea of thriving draws parallels between thriving people and thriving organizations; i.e., an organization cannot thrive if the people within it are not thriving. This framework identifies the complexity involved in cultural development and highlights "areas that are often neglected, especially by those who search for easy fixes and quick solutions" (p. 4). According to Tobias, a thriving person is one who has initiative, discipline, and accountability, invests in themself, is reflective and autonomous, and maintains a high regard for themself as well as others. Appendix A outlines each of these characteristics as Tobias defines them.

Using the definitions outlined in Appendix A and building on an existing survey also developed in the SenseMaker platform [28,29], the survey questions were created to guide inquiry and clarify interpretations of narratives provided by participants [19]. Given the unusual circumstances brought by COVID-19, exploring what it means to be a thriving person, or more specifically, a thriving student, is important to understand the role systems (the universities) played in minoritized students' ability to thrive during this time.

## 2. Materials and Methods

SenseMaker [19] is an online mixed methods data collection and analysis platform that involves four iterative steps: (1) *Initiation* is the process of designing signifiers, testing, and deploying the instrument; (2) *Story Collection* is the process of collecting data through narratives; (3) *Sensemaking* is the process of exploring and analyzing patterns of the collection of narratives; and (4) *Response* is the process of amplifying positive stories and dampening negative stories to nudge the system towards an adjacent possible with the goal of improving the system.

## 2.1. Step 1: Initiation

This study builds upon a previously developed signification framework based on the concept of thriving [20,27,30] and a similar study conducted at the University of Georgia [28]. This framework included five triads, three dyads, and six to eight multiple-choice questions outlined in Table 1.

**Table 1.** Five triads and three dyads that participants will use to self-signify their own stories.

| Theoretical Grounding | Question | Triad |
|---|---|---|
| Autonomy | My actions were motivated by . . . | Expectations of others, self-care, necessity |
| Investment/discipline | What was valued in this story was . . . | Willingness to experiment, grit and perseverance, planning and efficiency |
| Internal alignment/alignment with others | The experience I shared influenced my sense of . . . | Confidence, purpose, belonging |
| Openness/reflectiveness | Any decisions that were made in this story were influenced by . . . | Intuition, self-reflection, feedback from others |
| Flexibility | Thinking about the future, this story encourages me to . . . | Embrace risk, be willing to adapt, rely on familiar ways of working |
| Thriving (overarching concept) + accountability (sub-feature) | Dyad: This story was about . . . | Struggle–progress |
| Collaboration/competition, selflessness/selfishness, social contribution | Dyad: In this story I decided to prioritize . . . | Myself/my own self-care–needs/expectations of others |
| Support, rate of change | Dyad: In this story, change is happening . . . | Too slowly–too quickly |

The research team, in collaboration with Cognitive Edge, developed these triads and dyads by mapping the appropriate data collection points in SenseMaker to the concept of thriving [27,31], modifying language and existing survey questions based on results from previous studies [20,28], and adding additional questions in response to what was most relevant to the experiences due to COVID-19. The goal of the project was to explore the concept of thriving in minoritized engineering students nationwide and capture how COVID-19 has impacted this population.

When developing the dyads and triads for the SenseMaker platform, it is important that the options presented align with the prompt and create extremes given the context of the spectrum. For example, in the triad in Figure 1, one could answer the question of "My actions were motivated by" with a multiple-choice single-option response of necessity, expectations of others, or self-care. Participants would then move the center hexagon around the triad to indicate where they fall on the triad. However, the presentation of prompts with responses in dyads or triads enables the participant to respond to the prompts more holistically considering all factors involved in their response. Thus, from a data analysis viewpoint, we can use dyads and triads to understand the extent to which the options influenced the participant's response to the corresponding prompt.

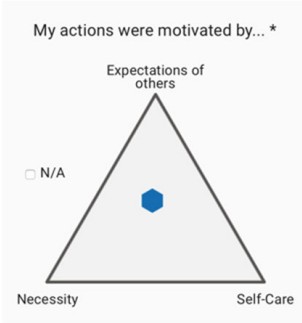

**Figure 1.** SenseMaker triad that probes the concept of autonomy. The hexagon can be moved within the boundaries of the triad. If the question is not applicable to the participant, they can select N/A.

*2.2. Step 2: Reflective Narrative Collection*

Narrative (i.e., data) collection entailed the distribution of a link to an online survey that poses the following question:

*"Imagine you are chatting with a friend or family member. Tell them about something you have experienced recently as an engineering student as a result of COVID-19."*

After submitting their narrative, participants were asked a series of questions about their story, which map to the triads and dyads in Table 1, such as the question shown in Figure 1.

This process is defined as self-signification and is one of the most unique and innovative features of SenseMaker studies. As opposed to researchers analyzing stories through their worldviews, participants make sense of their own stories through the provided signification framework, thus reducing researcher bias [19].

When presented with a triad, participants were asked to move a dot on the triangle to the position that best fits their story. In the analysis software, each dot represents a story that can be examined for trends.

*2.3. Step 3: Sensemaking "Explorative Pattern Analysis and Collective Sensemaking"*

Data collected included participants' narratives (qualitative) and responses to dyads, triads, and multiple-choice questions (quantitative data). Many participants engaged in the study in June and July 2020.

As data were collected, trends of the data were analyzed with the goal of detecting positive trends and weak signals that emerged. By identifying these trends, we are able to communicate the way in which faculty, staff, and administrators can make positive changes in their respective roles in order to "amplify the positive and dampen the negative" with the goal of creating "more stories like these, and fewer stories like those" [19]. By examining triad and dyad data (e.g., dots on triangles) as well as XY plots and heatmaps, we were able to determine an area of improvement in the system: self-care/mental health. In Figure 2, the circle in the upper right corner highlights the number of stories in which participants indicated prioritizing themselves and their mental health while also experiencing a lot of struggles during this time. Through this research, we hope to inspire changes that would shift these stories to the left as indicated by the arrow in Figure 2.

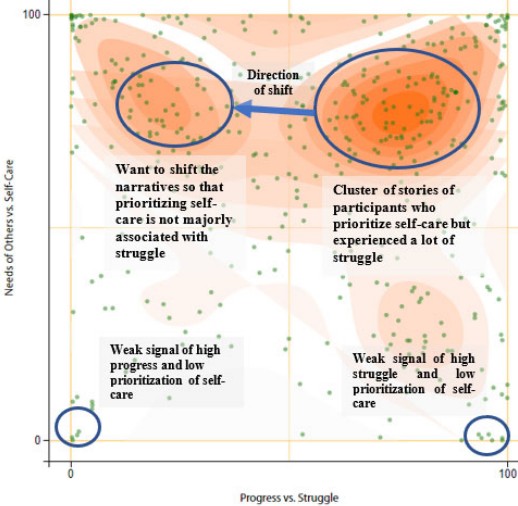

**Figure 2.** XY plot created from constructs from two dyads shown here as a heatmap. The small dots indicate participants.

As indicated by the color intensity of the heatmap, while there are some stories that prioritized mental health with little struggle, there are not nearly as many of those who indicated having a high struggle during this time. It is also worth noting the weak signals

in the bottom corners of Figure 2. Weak signals are areas that may signal opportunities for growth in the longer term.

A series of chi-squared tests and one-way ANOVAs were conducted to see if there was any statistical significance in the data. To conduct the one-way ANOVAs, the dyad of "Prioritizing Myself vs. Prioritizing the Needs/Expectations of Others" was maintained as a continuous dependent variable and evaluated for correlations for the following independent variables: (1) feelings about experience; (2) year in school; (3) gender identity; (4) racial identity; (5) family income; and (6) major. ANOVA tests were also conducted at the intersection of race and gender. To conduct the chi-squared tests, the average value for participants' responses was used as a threshold. If a participant's response was greater than the average, they were labeled as "Prioritized Self-Care/Mental Health". If a participant's response was lower than the average, they were labeled as "Did Not Prioritize Self-Care/Mental Health".

Participant narratives were analyzed using first-level and pattern coding methods developed by Miles et al. [31]. First-level coding was used for summarizing segments of data based on four keywords relating to mental health: mental health, depression/depressed, anxiety/anxious, and stress(ed). Once first-level coding was completed, pattern coding was conducted to identify themes relating to the mental health of participants during the pandemic.

### 2.4. Step 4: Response "Amplify the Positive and Dampen the Negative to Nudge the System to an Adjacent Possible"

Through continued data analysis, we have the tools to communicate to institutions, students, and the broader community the impacts COVID-19 has had on minoritized engineering students. There are several important themes that emerged from these data. For this paper, we are focusing specifically on the theme of self-care/mental health given that, during a time when many are suggesting that we take special care of mental well-being, minoritized students have not done so for a variety of reasons. In this paper, we aim to highlight the experiences of minoritized students' prioritization of mental health during this difficult time.

### 2.5. Validity and Reliability

Traditionally, the validity and reliability of a study, particularly as they pertain to mixed methods approaches, are based on the triangulation of measures. In surveys, this can include both open and closed-ended questions to measure the construct of interest [32]. Given the single prompt (open-ended question), it is important that when using the SenseMaker platform, each of the options on the triads and dyads (closed-ended questions) work together to maintain construct validity [33].

Another validation method is member checking which is defined as taking cleaned data and interpretations back to participants to confirm that an accurate account of the narrative was conducted [33]. According to Guba and Lincoln [34], member checking is the most important method of establishing credibility within a study. This process takes place in Step 3 (Section 2.3) which is the collective sensemaking process. Because of the anonymity of the survey, collective sensemaking may not include original survey participants but can include people who identify with survey participants (race, gender, year in school, etc.).

The reliability of a study is indicative of how consistent the instrument is in measuring constructs [35]. As mentioned previously, the SenseMaker survey was distributed was modified and enhanced based on a similar pilot survey distributed locally [20,28]. Modifications to the survey were conducted in collaboration with a Cognitive Edge team member (who also helped develop the pilot survey) and were performed to capture more accurate responses relevant to the pandemic.

### 2.6. Recruitment

There were 500 students (undergraduate and graduate) who participated in this study. Participants were required to be enrolled at a 4-year degree-granting institution and self-

identify as a minoritized student. Social media, personal connections, and listservs for the National Society of Black Engineers (NSBE) and the Society of Hispanic Professional Engineers (SHPE) were used in our recruitment methods. Participants received a USD 10 electronic Amazon gift card as a thank-you for their participation. Given the demographics of our participants, the research team opted not to include a "control group". Due to the sensitivity of the COVID-19 pandemic and the marginalization experienced by minoritized engineering students (prior to and ongoing since the pandemic), our goal is to not present a struggle Olympics amongst participants but simply present and center the experiences of our participants as they are valid without the need to be compared to a majority or "control" group.

## 3. Results

As previously mentioned, SenseMaker is a unique mixed methods tool that collects both qualitative and quantitative data from each participant. In this section, we will discuss how these methodologies work together to provide a better understanding of how minoritized engineering students prioritized their mental health.

### 3.1. Quantitative Findings

RQ1: How Many Minoritized Students Chose to Prioritize Their Mental Health?

Of the total number of stories shared, ~46% of participants indicated that they prioritized their mental health versus ~54% of participants who indicated prioritizing the needs of others. Appendix B details how we aggregated the data by overall experience (Table A2), year in school (Table A3), gender identity (Table A4), racial identity (Table A5), family income (Table A6), and major (Table A7).

Using a confidence interval of 95%, the results for each of the chi-squared tests that were conducted indicated that there were no statistical differences in the data. Additionally, the results for each of the one-way ANOVAs that were conducted also indicated no statistical significance in the data. While there were no statistical significances present when comparing the prioritization of self-care/mental health to the various independent variables, we recognize that people are more than just numbers and their experiences should not be minimized to the results of statistical tests. Therefore, in Section 3.2, we present the findings of our qualitative data collection.

### 3.2. Qualitative Findings

Using the four keywords relating to mental health, mental health, depression/depressed, anxiety/anxious, and stress(ed), eight major themes emerged from the data outlined in Table 2.

**Table 2.** Emergent themes related to mental health as experienced by participants.

| Theme | Definition: Participants Expressed . . . |
|---|---|
| Depression, Anxiety, and Stress | . . . experiencing increased depression, anxiety, and/or stress due to the pandemic. |
| Isolation | . . . experiencing increased loneliness and reduced access to community support. |
| Overwhelmed | . . . being overwhelmed by the rapid shift and increased responsibilities. |
| Home Life Struggles | . . . difficulty working/focusing at home, increased domestic responsibilities, and disputes with family. |
| Motivation | . . . experiencing reduced motivation to complete tasks. |
| Lack of Empathy | . . . experiencing a lack of care and support from instructors. |
| Coping | . . . finding things to help them escape their reality. |
| Support | . . . experiencing support amidst the challenges of the pandemic. |

Due to the permissions participants allowed for readers to view their narratives (everyone vs. researchers only), many specific examples that resulted in the emergent themes are unable to be shared in this paper. However, the narratives that are permissible for public viewership will be shared as participant profiles. Additionally, results are presented as profiles to demonstrate the depth of participants' experiences versus merely examples of emergent themes.

RQ2: How Do Minoritized Students Describe Their Experiences and Choices to Prioritize (or Not Prioritize) Their Mental Health?

Participant 1 Profile: Hispanic/Latino Cis-Woman, Engineering Education Graduate Student, Extremely Positive Feelings, Prioritized Mental Health/Self-Care

Story Title: Sow mental health, harvest healthy work environments.

*Before COVID-19 hit, I had just joined a different lab group. I had met all the members already and was participating in bi-weekly meetings with them. After COVID-19, the professor leading the group changed all our meetings to weekly optional check-ins. I was really surprised by it since I'd heard from other students that their advisors were giving them loads of work since they were home all the time now. Instead, my new advisor focused greatly on mental health and encouraged us to talk to each other when needed and to be open about our worries. He himself was very open with us and that helped us see that everyone is going through the same thing together. After each check-in, we decided how much work we were able to do for the week. He said we were better at assessing our own progress and capabilities than he was. I really appreciated that. Even when we didn't have a meeting scheduled, some of the lab members would tune in to the lab channel and keep each other company while working. Our advisor brought us together and gave us a space to be open about our concerns.*

Participant 2 Profile: White Cis-Woman, Civil Engineering Junior, Positive Feelings, Prioritized Mental Health/Self-Care

Story Title: Finally Able to Breathe.

*This experience has truly allowed me to take time to focus on my mental health and dwell on my options for my future after graduation, and this is something I have never really been able to think about before. I think the college environment, especially for engineers, can become excessively competitive and numbing. It is easy to lose sight of why you pursued your respective degree in the first place because you become so focused on getting a good GPA, staying active in clubs and organizations, balancing a social life, keeping physically active, etc. This was especially true right before classes were switched to online right after winter break. It was the middle of my second semester of junior year, and I wasn't able to reflect on my passions, experiences, and future because I was drowning in engineering coursework, extracurriculars, and work. However, once online classes started, I had more time to breathe and reflect. I wouldn't say I have a better focus than I did before everything started, but I was able to really appreciate my classes and realize that, at this point in my college career, I am learning things I will directly apply in my engineering career. I have also found a greater appreciation for my peers, professors, advisors, and the academic community as a whole. I have found my motivation and energy again, and I think from now on I will be comfortable in taking time to reflect and assess.*

As demonstrated by these participants' narratives, there were minoritized students who experienced Support amidst the challenges of the pandemic. Participants 1 and 2 had positive experiences during the pandemic and were able to prioritize their mental health largely due to the Support they received and shared through their narratives. While Participant 1 focused primarily on the Support they received from their graduate advisor, Participant 2 shared that their Support came from a variety of places such as peers, professors, and academic advisors. There were participants who indicated having extremely positive or positive

feelings about their experiences and indicated that they did not prioritize their mental health but made no mention of the subject in their narratives.

While many participants who had positive experiences indicated a strong priority of their mental health, that was not the case for participants who had less than positive experiences (Participants 3, 4, and 5).

Participant 3 Profile: Black/African American Cis-Woman, Chemical Engineering Senior, Neutral Feelings, Did Not Prioritize Mental Health/Self-Care

Story Title: Impacts of COVID from academia to personal life.

*As schools were closing down around us, it took [HBCU] days to figure out if we were going to go home or stay on campus. It was late Thursday when they decided that they were going to shut down for, "two weeks" and informing students on campus that they had two days to get off campus. I witnessed friends scurry to get plane tickets or figure out where they were going to stay before traveling home because finances did not allow them to travel home within 48 hrs. Professors and students had to transition rapidly to online learning. As an engineering student that is accustomed to being in a classroom or studying in the engineering lounge, it was difficult. First two weeks, I lost track of assignments due. Additionally, Professors in my humanities courses were adding on assignments because they thought we had the time. As a graduating senior, I had an extensive senior design project that needed more attention. I had to take my time management seriously by utilizing a chalk and dry erase board to write down assignments due. At the same time, my family and friends in New Jersey and New York were impacted by COVID the most. Hearing what they were going through and see how it was impacting them took a toll on my mental and emotional health. After the semester was over [HBCU] continued to show a lack of support towards their students as well as their professors. Recently, they gave students two weeks to head to the institution to pack up their dorms. Imagine students for the Midwest to the West Coast having a short amount of time to buy plane tickets to pack up their dorm. Some tickets were expensive. When the students and parents complained, they were told to "hire a private moving company". That is not a response that they should hear. Especially, when most students are going through a lot during the pandemic and might not have the finances. Additionally, being a graduate I was looking forward to starting my career as a chemical engineer at [company]. COVID has hindered my start date. I am a co-homeowner and have bills to pay and need to start working to not fall behind. The stimulus check provided was enough to cover 1 month and a half worth of bills. Nonetheless, being home has been a real struggle because I am used to working whether it is at a job or on assignments for school.*

Participant 4 Profile: Black/African American Cis-Woman, Computer Science Junior, Negative Feelings, Prioritized Mental Health/Self-Care

Story Title: Overcoming Hard times or Not.

*During the COVID-19 crisis, I experienced an overwhelming increase in anxiety. Right before quarantine started, I was already experiencing a lot of anxiety attacks and my mental health was going in the opposite direction of being healthy. This caused me to take a trip to the hospital which was ordered by my school counselor. I wasn't fully recovered or at least good enough for something like this to happen to me. Once I realized that everything was closing and I had no outlet or get away from my problems, things went left. I have never felt so confined to just my apartment and making store trips which wasn't doing much for me. It started to affect my work on top of some teachers not being able to transition as easily as other because of their own problems at home. This had to literally be the worst time of my life. I was forced to be by myself, especially if I wanted to stay healthy. I spent everyday doing homework and studying just thinking, I can't wait until it's the summer and school is over but even that overwhelmed me. Once the summertime came, then what? I had no idea how this virus was going to turn out. It caused me to just get by in school. I wasn't focused one bit because everyday was*

*a challenge. Luckily to say that I did finish with some pretty good grades, but it took everything I had in me. I've never pushed so hard in my life and I have been through and got through a lot. Still today, I am dealing with some anxiety but at least I don't have school to add to it. The only thing I can do is receive support from understanding people and continue to take care of myself.*

Participant 5 Profile: Black/African American Cis-Man, Chemical Engineering Senior, Extremely Negative Feelings, Did Not Prioritize Mental Health/Self-Care

Story Title: The Pandemic of Ignorance.

*Being an engineering student during the COVID-19 crisis was extremely stressful and tiring. At my school, we were required to attend double the amount of lectures in addition to the live lectures. This was inconvenient for me because something I think the professors still fail to understand is that even though we are at home, we do not have more time. As time progressed, the pre-recorded lectures got worse as professors completely skipped examples and just told us to work on them ourselves. This completely defeated the purpose of having a pre-recorded lecture if we weren't doing problems. Additionally, we were assigned a ridiculous amount of work in the courses. Students were barely given any extra time to submit assignments and had to deal with the stress of penalties from the professor if anything was submitted a minute late. The submission portals disappeared at the exact time they were due (ex. 10:30:00) so if you attempted to submit even a second later, it was gone. There was no consideration for technical difficulties and it required me to rush to submit my assignments rather than check my work. When I tried to respectfully tell professors the issues that I experienced, they tended to defend themselves and disagree with my experiences. I believe it is important as students for our opinions to be taken into consideration and not brushed aside. Since we have been home, the professors have not been considerate of the overlapping schedules of our other chemical engineering courses. If we were in person, most professors would try to make sure our assignments and exams don't overlap but now it seems as if they do not care. I would have multiple weeks where I had three exams, multiple quizzes, and multiple homework assignments. This was just in the chemical engineering courses. Students have family responsibilities, research responsibilities, and clubs and organizations that they are a part of that continue to meet throughout the quarantine. It is very disturbing and disheartening to see how our professors have handled some of our concerns in these courses. I know that this is not an ideal situation for anyone to be in, but I find it hard to even prioritize my physical and mental wellbeing because my life is completely consumed by schoolwork. For many students, we are trapped inside our rooms. Although the junior year curriculum is known as the worst year of the chemical engineering major, I feel as if the professors can be more accommodating to student, concerns. Since I have been a part of the department at my school, the difficulties and hardships that students face are treated more as a rite of passage than serious issues that faculty try to mitigate. This major is already difficult enough and especially when you feel as if you do not have people to advocate for you in the major or a strong support system within the department, this can make it that much harder. Additionally, I thought that the P/F option was extremely necessary as we are in a global pandemic. It seems absurd that instead of focusing on our physical and mental wellbeing, our peace is being disturbed at home and we are forced to wonder if we will or will not pass our courses. I fully understand believing that students should progress to the next course after mastery of the material but penalizing students for situations that are out of their control and forcing them to stay an extra year at school is extremely unfair and inconsiderate. It is impossible to imagine the different scenarios that every student is facing and what their home life is like. Many students are currently housing insecure, food insecure, and without income due to school closures. In addition, there are students taking care of children or elderly folks, students without internet access, students living in abusive households, and students with disabilities without access to accommodations. Finally,*

*some students will inevitably become ill with COVID-19. It is unreasonable to maintain the expectation that students uphold their grades while grappling with these difficulties.*

For participants who indicated having less than positive experiences, prioritization of mental health varied in no particular order. Participant 3's narrative is an example of a student who was Overwhelmed during the pandemic and experienced a Lack of Empathy, Depression, Anxiety, and Stress, and Home Life Struggles during the pandemic. Some of the stories were like Participant 4's in which having a negative experience prompted them to focus more on their own well-being versus meeting the needs or expectations of others, but they still experienced Depression, Anxiety, and Stress, Isolation, and Support during the pandemic. Other stories were like Participant 5's in which having a negative experience is exactly the reason why they found it difficult to prioritize their mental health. The pressure to manage responsibilities during this unprecedented time (and some somewhat urgently) further exacerbated the de-prioritization of mental health and self-care and is another example of those who were Overwhelmed and experienced a Lack of Empathy, Isolation, and Home Life Struggles.

## 4. Discussion

Minoritized students are not a monolith [36–39]. Of the students who did not prioritize their mental health, some had bad experiences while others had positive experiences. Both can serve as examples for institutional leaders who are working to make improvements. When connecting the findings back to Aston's Theory of Student Involvement, the qualitative and quantitative results clearly highlight the impact COVID-19 has had on minoritized students in engineering. Most evidently, their environments were impacted by being uprooted from traditional classrooms and losing direct contact with faculty members, classmates, and other hands-on support systems [40]. Studies have found that the correlation between mental health, belonging, and persistence in engineering is very strong. Individuals will disengage from situations where they feel a threat to their abilities based on negative stereotypes and experiences [41]. Within this study, students elaborated on their long-term concerns for their grades and the stress it has caused, and research continues to highlight these challenges and how they can be addressed. It is also worth noting that some concerns expressed by participants, such as a lack of empathy for minoritized student experiences, were present prior to COVID-19, and the pandemic further exacerbated these issues [42–50].

At the classroom level, several institutions are incorporating reflection periods that are similar to the journaling process. Recent studies are showing that these types of activities promote community and show students that faculty care about mental health and how it affects students' academic success [51]. Students are able to anonymously place symbols that represent their feelings, allowing for both the professor and student peers to obtain an understanding of where everyone was emotionally. This type of communication and rapport is needed, especially within virtual settings, and also to bridge the gap between diverse students who are dealing with diverse challenges during this time and who would otherwise not relate to one another [51]. At the University of California at Davis, an elective seminar was offered to support students and resulted in positive feedback including feeling appreciated, in general, and thankful for the advice around time-management topics, self-care, and wellness [16].

Within this discussion, we have focused on solutions given the importance of this topic and extensive literature around how concerns arise related to self-care and mental health for all college students. Administrators and leaders at the various levels of higher education will benefit from investing time and resources into better understanding students and their self-care, especially when it comes to mental health. Overall, future research will need to explore and understand the short-to-long-term implications of COVID-19 for this population, especially as uncertainty around the pandemic continues.

*Limitations*

While an extensive amount of quality student responses was collected for this project, it did not come without limitations. When considering that the target population focused

on minoritized students, a higher number of Native American and other minority students would have been ideal. When considering the national engineering enrollment rates, our findings did have an imbalance of representation across the groups. A higher representation of students from Tribal Colleges and community colleges would have added additional insight to our findings. Given the state of the pandemic and various data collection methods that have been disbursed to students, survey fatigue is acknowledged. This goes along with the timing of the survey. Most students are not on campus and, therefore, are not checking their email accounts as often, thus decreasing the number of survey respondents.

Another limitation of this project was not implementing specific components of the SenseMaker tool. Ideally, data collection is an ongoing, iterative process in which researchers continuously gain feedback from participants as they work to gradually shift narratives in a positive direction [19]. However, we have learned that it is difficult to implement Step 4, the process of using responses to amplify the positive and dampen the negative in a continuous feedback loop, of the sensemaking process with a nationwide data pool. This is not due to the size of our dataset but rather the level of direct influence we have over changing the specific experiences of those who participated in the study. While we are attempting to mimic this process through public webinars and seminars, we are unable to control for whether the information we are disseminating based on our findings is actually being used to make positive shifts in the narratives.

## 5. Conclusions

The COVID-19 pandemic has had an unprecedented impact on higher education institutions. The well-being of students from historically minoritized groups within engineering education was of great concern due to the challenges they faced even before the pandemic. Unique findings around self-care and mental health from initial analysis led to a focus on this phenomenon, given the heightened focus on the subject during this time. A key finding was that, even within the minoritized engineering population, student respondent decisions to prioritize their mental health were not monolithic. There were students who had positive experiences during this time which resulted in a strong prioritization of self-care, while others had negative experiences which resulted in a varied prioritization of self-care. Several students chose to prioritize their mental health as a way to remain mentally and emotionally afloat during this time. For others, their negative experiences resulted in them abandoning self-care strategies in order to tend to the needs of academics or family members. When it comes to this engineering student population, the types of sociological capital are dynamic in how they interact with one another. Overall, the topics of self-care and mental health for engineering students should be prioritized by higher education leaders, especially during and after the COVID-19 pandemic.

**Author Contributions:** Conceptualization, R.S.L., T.F. and Z.V.S.; methodology, R.S.L. and Z.V.S.; software, R.S.L., D.A. and Z.V.S.; validation, R.S.L. and T.F.; formal analysis, R.S.L. and Z.V.S.; investigation, R.S.L. and T.F.; resources, R.S.L.; data curation, R.S.L.; writing—original draft preparation, R.S.L., T.F. and D.A.; writing—review and editing, R.S.L., A.P. and Z.V.S.; visualization, R.S.L.; supervision, R.S.L.; project administration, R.S.L.; funding acquisition, R.S.L. and T.F. All authors have read and agreed to the published version of the manuscript.

**Funding:** This research was funded by the National Science Foundation grant number 2029564.

**Institutional Review Board Statement:** The study was conducted in accordance with the Declaration of Helsinki and approved by the Institutional Review Board (or Ethics Committee) of the University of Georgia (protocol code PROJECT00002167 and 16 April 2020).

**Informed Consent Statement:** Informed consent was obtained from all subjects involved in the study.

**Data Availability Statement:** Not applicable.

**Acknowledgments:** We would like to acknowledge the support of Nicki Sochacka for introducing Sensemaker to the research team.

**Conflicts of Interest:** The funders had no role in the design of the study; in the collection, analyses, or interpretation of data; in the writing of the manuscript; or in the decision to publish the results.

## Appendix A  Definitions of a Thriving Person

**Table A1.** Definitions of the thriving person (Tobias, 2004, Appendix).

| Concept | Definition |
|---|---|
| Initiative | Is initiative-taking, proactive, industrious, goal-oriented, decisive, impactful, purposeful, forward-moving; confronts obstacles; is ready to respond; does not exhibit withdrawal, fatalism, or helpless resignation in the face of barriers |
| Discipline | Is disciplined, persistent, effortful, planful, reliable; orders life and priorities; constructively uses systems, structures, schedules, and methods to foster accomplishment; exercises self-control; meets or fulfills obligations and commitments; stays within appropriate boundaries; stays focused and appropriately on track without undue distractions or impulsiveness; avoids the easy way out if that diminishes effectiveness; is efficient in use of time and resources |
| Accountability | Holds self widely accountable; maintains high standards and a continuous growing edge; is self-demanding; takes responsibility for self-improvement; measures own progress; acknowledges failures and is not content with excuses; accepts both credit and blame; constructively compares self with others and faces up to criticisms and weaknesses; does not just wait for issues to surface but attempts to anticipate the unspecifiable and to prepare for adversity or opportunity |
| Investment | Invests in self; prepares for the future; delays the gratification of short-term goals in the service of long-term goals; balances priorities and goals to optimize eventual effectiveness; husbands resources; maintains reserves; ministers to self; builds on native assets; avoids squandering resources, talents, time, or energy because of shortsightedness, greed, or lack of perspective |
| Openness/Reflectiveness | Is open to ideas, feelings, feedback, experience; is inquiring, progressive, receptive to divergent perspectives; is skilled in resolving differences of viewpoint; is not dogmatic; willingly questions tradition and the status quo; is tolerant of ambiguity, inconsistencies, contradictions; is open to humor; is self-scrutinizing, reflective, willing to face up to issues, willing to acknowledge faults or failures; seeks out own prejudices and biases; invites diverse input; is willing to test conventional reality; searches for own biases and blind spots; is receptive to constructive criticism; is sensitive to feelings, empathic; has a good sense of own strengths and shortcomings; searches for discrepancies between ideals and actions |
| Flexibility | Is flexible; seeks personal growth; adapts to new environments, challenges, crises, internal conflicts, changes; is resourceful; is able to change course; does not over persist or get stuck in blind alleys; tries new ways around resistant obstacles; is willing to risk and to try new things; embraces change; welcomes novelty; is able to change own mind |
| Autonomy | Is autonomous, weaned, independent, self-reliant; thinks for self; has courage of convictions; has clear sense of self-identity, self-direction, internal rudder; can handle rejection; can go against the flow when appropriate; appropriately conforms, but is not conformist; is willing and able to make own choices and decisions; is able to assert own needs and priorities; can find own structure amidst unclear pathways |
| Alignment with Others | Is interrelated, aligned with others; exhibits social consciousness; feels ownership/responsibility for the welfare of an increasingly widening circle of others; reaches out to others; is able to join and fit in; is able to contribute as a member of a team; is tolerant, accepting, responsive, affirming, approachable, communicative, capable of intimacy and trust; has com-passionate regard for others; is capable of forgiveness; has a sense of intergenerational reciprocity; respects personal boundaries; attempts to work out differences and reach compromises; has brotherly/sisterly feelings toward peers and parental feelings toward the community; builds willing constituencies of mentors, peers, and followers |
| Internal Alignment | Is internally aligned, personally integrated, genuine, authentic, transparent; has a well-developed and well-articulated self-concept and value system that are consistent with feelings and behavior; stands for something; has a unifying philosophy and sense of purpose/mission; has vitality, zest/appreciation for life; has integrity and an ethical sense; is self-aware, insightful; is able to resolve internal conflicts; has humility and self-esteem; is realistic, objective, able to adopt a balanced perspective; exhibits a minimum of hypocrisy |

## Appendix B  Quantitative Data Tables

**Table A2.** The number of students who did or did not prioritize self-care/mental health according to how they felt about their experience.

| How Do You Feel about Your Experience? | Did Prioritize (C) (R) | Did Not Prioritize (C) (R) | Total |
|---|---|---|---|
| (1) Extremely Positive | 16 (6.9%) (53.3%) | 14 (5.2%) (46.7%) | 30 |
| (2) Positive | 58 (25%) (57.4%) | 43 (16%) (42.6%) | 101 |
| (3) Neutral | 46 (19.8%) (38.7%) | 73 (27.2%) (61.3%) | 119 |
| (4) Negative | 93 (40.1%) (48.2%) | 100 (37.3%) (51.8%) | 193 |
| (5) Extremely Negative | 19 (8.2%) (37.3%) | 32 (11.9%) (62.7%) | 51 |
| (6) Prefer not to answer | | 6 (2.2%) (100%) | 6 |
| Grand Total | 232 | 268 | 500 |

**Table A3.** Number of students who did or did not prioritize self-care/mental health by year in school.

| Year in School | Did Prioritize (C) (R) | Did Not Prioritize (C) (R) | Total |
|---|---|---|---|
| (1) Freshman | 49 (21.1%) (51%) | 47 (17.5%) (49%) | 96 |
| (2) Sophomore | 43 (18.5%) (44.8%) | 53 (19.8%) (55.2%) | 96 |
| (3) Junior | 61 (26.3%) (45.5%) | 73 (27.2%) (54.5%) | 134 |
| (4) Senior | 50 (21.6%) (45.9%) | 59 (22%) (54.1%) | 109 |
| (5) Graduate Student | 27 (11.6%) (49.1%) | 28 (10.4%) (50.9%) | 55 |
| (6) Prefer not to answer | 2 (0.9%) (20%) | 8 (3%) (80%) | 10 |
| Grand Total | 232 | 268 | 500 |

**Table A4.** Number of students who did or did not prioritize self-care/mental health by gender identity.

| Gender Identity | Did Prioritize (C) (R) | Did Not Prioritize (C) (R) | Total |
|---|---|---|---|
| Cis-Man | 89 (38.4%) (53.3%) | 78 (29.1%) (46.7%) | 167 |
| Cis-Woman | 124 (53.4%) (42.2%) | 170 (63.4%) (57.8%) | 294 |
| Non-Binary/Non-Conforming | 2 (0.9%) (66.7%) | 1 (0.4%) (33.3%) | 3 |
| Other gender expression | 4 (1.7%) (33.3%) | 8 (3%) (66.7%) | 12 |
| Prefer not to answer | 13 (5.6%) (56.5%) | 10 (3.7%) (43.5%) | 23 |
| Transgender Man | (0%) (0%) | 1 (0.4%) (100%) | 1 |
| Grand Total | 232 | 268 | 500 |

**Table A5.** Number of students who did or did not prioritize self-care/mental health by racial identity.

| Racial Identity | Did Prioritize (C) (R) | Did Not Prioritize (C) (R) | Total |
|---|---|---|---|
| American Indian/Alaskan Native | 1 (0.4%) (100%) | (0%) (0%) | 1 |
| Asian | 23 (9.9%) (44.2%) | 29 (10.8%) (55.8%) | 52 |
| Black/African American | 112 (48.3%) (49.6%) | 114 (42.5%) (50.4%) | 226 |
| Hispanic/Latino | 16 (6.9%) (48.5%) | 17 (6.3%) (51.5%) | 33 |
| Mixed Race | 23 (9.9%) (46.9%) | 26 (9.7%) (53.1%) | 49 |
| Native Hawaiian/Other Pacific Islander | 1 (0.4%) (100%) | (0%) (0%) | 1 |
| Other | (0%) (0%) | 1 (0.4%) (100%) | 1 |
| Prefer not to answer | 2 (0.9%) (28.6%) | 5 (1.9%) (71.4%) | 7 |
| White | 54 (23.3%) (41.5%) | 76 (28.4%) (58.5%) | 130 |
| Grand Total | 232 | 268 | 500 |

**Table A6.** Number of students who did or did not prioritize self-care/mental health by family income.

| 2019 Family Income | Did Prioritize (C) (R) | Did Not Prioritize (C) (R) | Total |
|---|---|---|---|
| (1) Less than USD 25,000 | 28 (12.1%) (48.3%) | 30 (11.2%) (51.7%) | 58 |
| (2) USD 25,000–USD 50,000 | 48 (20.7%) (52.2%) | 44 (16.4%) (47.8%) | 92 |
| (3) USD 50,000–USD 100,000 | 54 (23.3%) (46.2%) | 63 (23.5%) (53.8%) | 117 |
| (4) USD 100,000–USD 200,000 | 43 (18.5%) (40.2%) | 64 (23.9%) (59.8%) | 107 |
| (5) More than USD 200,000 | 11 (4.7%) (31.4%) | 24 (9%) (68.6%) | 35 |
| (6) Prefer not to answer | 48 (20.7%) (52.7%) | 43 (16%) (47.3%) | 91 |
| Grand Total | 232 | 268 | 500 |

**Table A7.** Number of students who did or did not prioritize self-care/mental health by major.

| Major | Did Prioritize (C) (R) | Did Not Prioritize (C) (R) | Total |
|---|---|---|---|
| Aeronautical and Astronautical Engineering | 3 (1.3%) (50%) | 3 (1.1%) (50%) | 6 |
| Aerospace Engineering | 11 (4.7%) (47.8%) | 12 (4.5%) (52.2%) | 23 |
| Biomedical/Biochemical Engineering | 19 (8.2%) (33.3%) | 38 (14.2%) (66.7%) | 57 |
| Chemical Engineering | 17 (7.3%) (39.5%) | 26 (9.7%) (60.5%) | 43 |
| Civil Engineering | 24 (10.3%) (43.6%) | 31 (11.6%) (56.4%) | 55 |
| Computer Science | 44 (19%) (61.1%) | 28 (10.4%) (38.9%) | 72 |
| Construction Engineering and Management | 3 (1.3%) (75%) | 1 (0.4%) (25%) | 4 |
| Electrical and Computer Engineering | 24 (10.3%) (47.1%) | 27 (10.1%) (52.9%) | 51 |
| Engineering Education | 2 (0.9%) (66.7%) | 1 (0.4%) (33.3%) | 3 |
| Engineering Science and Mechanics | (0%) (0%) | 1 (0.4%) (100%) | 1 |
| Engineering Technology | 2 (0.9%) (40%) | 3 (1.1%) (60%) | 5 |
| Environmental Engineering | 3 (1.3%) (50%) | 3 (1.1%) (50%) | 6 |
| Food, Agricultural, and Biological Engineering | 2 (0.9%) (33.3%) | 4 (1.5%) (66.7%) | 6 |
| Industrial and Systems Engineering | 20 (8.6%) (48.8%) | 21 (7.8%) (51.2%) | 41 |
| Materials Science and Engineering | 7 (3%) (53.8%) | 6 (2.2%) (46.2%) | 13 |
| Mechanical Engineering | 43 (18.5%) (50%) | 43 (16%) (50%) | 86 |
| Mining and Materials Engineering | 1 (0.4%) (20%) | 4 (1.5%) (80%) | 5 |
| Ocean Engineering | (0%) (0%) | 2 (0.7%) (100%) | 2 |
| Other | 6 (2.6%) (30%) | 14 (5.2%) (70%) | 20 |
| Welding Engineering | 1 (0.4%) (100%) | (0%) (0%) | 1 |
| Grand Total | 232 | 268 | 500 |

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
