# Peer review of "Using SenseMaker® to Understand the Prioritization of Self-Care and Mental Health of Minoritized Engineering Students during the 2020 Global Pandemic"

_education, doi:10.3390/educsci13070643_

Round 1
Reviewer 1 Report
Unfortunately, despite the relevance and astuteness of this essay, there are problems with grammar, word choice and transitions throughout the essays so that literature review is a clunky read.
Also if you wish to present results despite the quant data showing no significant differences, as a trend, please give more than one example.
The method was explained well, but the lit review, results were not written in a way that is sufficiently compelling. . And it is certainly compelling research.
I would advise getting some editorial help with wording and work to fill out less sturdy sections while methods can remain pretty much the same.
The summaries of paragraphs are too general and a bit repetitive forestalling the development of the argument. The reader is also jarred by sentences that seem placed in the wrong paragraph.
I hope these changes which ask for filling out more material and important but editing changes can be accomplished as the work with URMs is really important and deserve wide readership.
Author Response
Thank you for your willingness to review and provide feedback for our paper. We have done some extensive work to improve the grammar of the paper and the results of the paper and hopefully create a more cohesive narrative based on you feedback.
Reviewer 2 Report
I enjoyed reading your manuscript.
Exploring the self-care and mental health of underrepresented engineering students during the pandemic is both timely and very important.
You do a terrific job of framing the issue of self-care and mental health challenges among these students. Your use of SenseMaker is also an interesting take on what might be a more traditional study that relies on a survey and interviews.
You managed to gather a rich store of data and lifted a number of representative quotes. I wonder if revisiting your data might add a bit more to your discussion and conclusion, because I believe you will find other commonalities or themes that are worth highlighting for readers.
In addition, building on your recommendations may help readers see how your findings offer insight into their particular contexts and ways forward in addressing these issues.
Finally, considering how your findings inform our understanding of self-care and mental health of underrepresented engineering students beyond the current pandemic will demonstrate to readers that your study is not restricted to a particular period of time. As you rightly suggest, supporting these particular students moving forward is essential if engineering education is going to address challenges that pre-date COVID-19.
Please see your manuscript and a number of edits and comments I made. I hope you find these useful.
I wish you the best with your paper!

Author Response
Thank you for your willingness to review and provide feedback for our paper. Your support and feedback were invaluable and we hope that you are pleased with our changes.
Reviewer 3 Report
The ms addresses an interesting topic, how underrepresented engineering students prioritize their self-care and mental health during the 2020 global pandemic. The study introduces a new tool for collecting data, the SenseMaker, and it is based on an extensive data. However, there are several shortcomings, which suggest to me that the ms is not suitable for publication in its current form. I hope that my comments below will help the author(s) to continue the work.
First of all, it seems that the SenseMaker tool is in focus on the ms. Its properties are described in detail, even if what can be done with it is not clearly relevant in terms of the study. At least it seems to be so for the reader. I suggest shifting the focus to the data and what can be concluded based on it.
The ms is unfortunately not very coherent and understandable. It is not obvious to the reader what is the relevance of 1.1 Background between Introduction and Literature review. Moreover, what is the rationale to have 1.3 Literature review and 1.4 Theoretical framework separated? Is not theoretical framework part of the literature review? On lines 55-56, it is maintained that “…our results found that very few…” Does this refer to the current study? Would it be more appropriate to refer to the results in the discussion, not in the introduction?
I suggest having a brief introduction, then literature review (including the theoretical framework) and after it the research questions. Now the RQs are presented after the background, but would it be more logical to present them after the literature review? At least it is the traditional way to structure research reports. Hence, I suggest rewriting the introduction, to think of what the main concepts are and define them, for instance self-care (line 53), mental health (line 91), stress culture (line 109), student involvement, and engagement (line 155).
Concerning the research questions, the first one (How many underrepresented students…) sounds quite technical to me. How many from what population? Why is this an important question? In general, the research questions should be derived based on the previous studies, so please argue how your RQs are based on what we know already about the topic and what gap they might fill in the research domain.
I also recommend presenting the data (2.6) in the beginning of materials and methods –section. Please, describe in detail the data collection procedure (how the participants responded, how long it took etc.).
Please, report in detail how the steps of the SenseMaker were conducted in the study. The dyads and triads came quite unexpectedly, it was not obvious if all of them were used in the study. If the struggle - progress dyad or expectations of others vs. self-care (autonomy) were crucial in the analysis, they should be presented also in the theoretical part. Please, clarify how the results of the signification framework are used.
Please, elaborate what is the quantitative data (multiple choice questions) referred to on line 238. What was asked and why? Figure 2 is interesting, but it is almost impossible for the reader to understand its meaning. I think that this requires more explanation: what means the direction of the shift; is it something the system did? How is struggle related to self-care? To what progress does it refer to? Is it linked to self-care? What are weak signals and why they may signal opportunity for growth? How are they related to the aim of the study? These should be elaborated in the introduction section. If you don’t use the results obtained from the heatmaps, consider is the Figure at all relevant to your study.
I would discuss validity and the reliability of the study in the discussion, not in methods.
In the Tables B1-B6, please explain what does C and R mean. The proportion of the participants in some cells is very low (e.g only one), so if you want to use the Tables, I suggest aggregating some of the groups.
In terms of the qualitative data, I think that it is not appropriate to report the stories in so detailed way. Would it be possible to make some summary of them? Even if the sample is big, it could be possible to identify some participants based on the stories (if e.g some peer knows that the person has participated). In addition, I don’t think that presenting some random examples will add any value to finding answers to the research question 2. I missed some analysis of the stories. Now the results are just describing what the respondents wrote. In addition, I did not find any link to the results presented in the triads and in the heatmaps.
It seems that there are no really relevant results obtained from the study; no differences in the quantitative data and conclusions based on the stories that experiences could be either positive or negative. This is not really a surprise. Thus, analyzing the data, I suggest you can focus on the stories, classify them to different categories and then look at for instance how self-care and needs of others are presented in them. Self-compassion is topical issue in educational sciences. I think that you might have a rich data if you just find some new and more intriguing perspective to it than presented in the ms.
Author Response
We present our research questions prior to the literature review because we did not want the priority of the paper to be buried too far from the introduction. Our literature review then supports why our research questions are important and thus the contributions we hope to make with this paper.
Thank you for your willingness to review and provide feedback for our paper. We have done some extensive work to improve the lit review and results of the paper and hopefully create a more cohesive narrative based on the feedback of the reviewers.
Round 2
Reviewer 1 Report
there are a few grammatical errors yet to be addressed, typos in the main, and there is one place where the transition needs to be fleshed out a bit more:
272- incomplete sentence
348-352- transition needed "Results (351) the single sentence para. seems out of place.
Passages from participant 1 and 2 do not seem to express support in the same way, is clarification possible?
428: lower case Professors needed
599: types of capital?
I would like the authors to be braver. They took up a very important topic (btw a control group would have been interesting). There are more strains of thought and reflections that could be elaborated from themes of overwhelmed, lack of empathy, isolation, home life struggles. Are any of these dimensions correlated or established in narrative or empirical studies as a greater tax on urm and women in engineering at grad or ungrad levels. This vein of copious research could be tapped to deepen the experiences of these students: if they were experienced some marginalization before, or had greater financial hardships, or were expected to do more at home (all of which have been researched with respect to STEM and students), does this indeed suggest that the intensification of these experiences during COVID could be even more impactful: the stretch would not be too tenuous if the exact or nearly exact same concerns were quite salient before COVID, if you find comparable students. This is not nto suggest to fly off into speculative space but to connect some themes in the research literature to make your argument more forceful. the airplane flies a little low.
Given recent controversy about closing schools during COVID (the cost benefit analysis of closing down live classrooms), and its differential deleterious effects on certain categories of students, I think the article is very important. I appreciate your caution in reporting your findings, but can you connect to other significant finding regarding experiences of minoritized students in engineering.
Author Response
Typos have been addressed.
Participants 1 and 2 do express receiving positive support but where they received it from was further explained in the updated manuscript.
Language about the decision to not use a control group was added to the manuscript. Our goal is to center the experiences of minoritized students not portray a competition of whose struggle was worse. It is oppressive to see minoritized experiences as valid only when they are compared to some control/majority group (because what would a control group truly be in this case when everyone experienced a global pandemic?).
Some additional literature was added to the discussion to link this manuscript to previous discussions about the struggles of minoritized students.
Reviewer 3 Report
Dear authors,
thank you for revising the ms. In general, the ms is now more coherent and well argued than the previous version. One small comment: In the materials and methods sections the SenseMaker tool is presented. It would be helpful if the rational behind the triads and dyads is clarified and elaborated. They were still unclear to me based on the theoretical framework used to develop the SenseMaker Platform.
Author Response
The following paragraph was added to the manuscript in section 2.1 to provide additional clarity.
When developing the dyads and triads for the SenseMaker platform, it is important that the options presented align with the prompt and create extremes given the context of the spectrum. For example, in the triad in Figure 1, one could answer the question of “My actions were motivated by” with a multiple choice single option response of necessity, expectations of others, or self-care. However, the presentation of prompts with responses in dyads or triads enables the participant to respond to the prompts more holistically considering all factors involved in their response. Thus, from a data analysis viewpoint, we can use dyads and triads to understand the extent to which the options influenced the participant’s response to the corresponding prompt.